# Haemophilia and Cancer: A Literature Review

**DOI:** 10.3390/jcm13061770

**Published:** 2024-03-19

**Authors:** Ezio Zanon, Annamaria Porreca, Paolo Simioni

**Affiliations:** 1Haemophilia Center, Internal Medicine—Department of Medicine, Padua University Hospital, 35128 Padua, Italy; paolo.simioni@unipd.it; 2Laboratory of Biostatistics, Department of Medical, Oral and Biotechnological Sciences, “G.d’Annunzio” University of Chieti-Pescara, Via dei Vestini 31, 66100 Chieti, Italy; annamaria.porreca@unich.it

**Keywords:** haematology, haemophilia, cancer, malignancy, congenital bleeding disorders

## Abstract

Background: Opinions in the literature on the impact of cancer on patients with haemophilia are contradictory. There is a lack of data on the clinical presentation and management of cancer in patients with haemophilia (PWH). Methods: Papers were found following a comprehensive search in PubMed, Google Scholar, and Scopus using the terms “cancer” and “haemophilia” without time limits and using the English language as a filter. The references from all the retrieved original articles and reviews were assessed for additional relevant articles. Results: The emergence of malignancies is one of the important causes of morbidity and mortality in PWH. In the past decade, the literature mainly focused on the epidemiology and outcome of blood-borne cancers in the haemophilia patient group, as the incidence of hepatitis B virus (HBV), hepatitis C (HCV), and HIV infection were high among them. However, with the introduction of recombinant clotting factor concentrates (CFCs), physicians now pay attention to non-virus-related malignancies. Bleeding and thrombotic complications are important causes of morbidity and mortality in critically ill patients with cancer; replacement therapy with factor VIII or IX or others should be maintained during antitumour treatment. Conclusion: Overall, managing cancer in patients with haemophilia requires careful evaluation and individualised planning involving a multidisciplinary team of physicians experienced in haematology, oncology, and surgery.

## 1. Introduction

Can congenital bleeding disorders and cancer be related in various ways? The scientific literature suggests that the relationship between congenital bleeding disorders and cancer is complex and multifactorial. Some congenital bleeding disorders, such as haemophilia, can increase the risk of developing certain types of cancer [1]. In other bleeding disorders such as von Willebrand disease (VWD), solid tumours and lymphoproliferative disorders are reported in many cases. However, the association between cancer and VWD is not well-established [2,3].

Additionally, patients with congenital bleeding disorders may undergo haemostatic therapies that, in turn, can influence the risk of developing tumours. Abnormal bleeding occurs in approximately 10% of solid tumour patients and a higher proportion of patients with hematologic malignancies [4,5]. Bleeding in cancer patients is associated with significant mortality, and there are various potential causes of new-onset bleeding in patients with malignancy [6]. These include pre-existing mild congenital haemophilia and other coagulation factor deficiencies: congenital von Willebrand disease, thrombocytopenia, decreased synthesis of coagulation factors due to liver dysfunction or vitamin K deficiency, tumour erosion through a vessel wall, disseminated intravascular coagulation (DIC), and acquired inhibitors (neutralising autoantibodies). The largest study on the epidemiology of cancer in patients with inherited bleeding disorders was published in 1979 by Forman [7]. Most of the patients in the study were affected by Haemophilia A or B. Haemophilia A and B are uncommon congenital coagulation disorders associated with an X-linked inheritance pattern. Haemophilia A results from a deficiency of factor VIII (FVIII), while Haemophilia B is caused by a deficiency of factor IX (FIX). In cases of severe haemophilia (FVIII or FIX < 1 international unit [IU]/dL), spontaneous or post-traumatic bleeding occurs, primarily affecting joints and other tissues. Some of these bleeding episodes can be life-threatening or pose a risk to organs [8,9,10]. The predominant morbidity is attributed to recurrent haemarthroses, leading to the development of a degenerative joint disease known as haemophilic arthropathy. This condition results in chronic pain and functional impairment [11]. Over the past few decades, the central approach to managing these disorders has been the prevention of bleeding episodes through replacement therapy, aiming to decrease both mortality and the onset of chronic arthropathy [9,11]. The issue of malignancies in patients with haemophilia is particularly intriguing because of the presumed protective effect of this hereditary bleeding disorder against cancer spread and dissemination [12].

This study aims to update the existing literature concerning the relationship between congenital bleeding disorders and cancer.

Most of the evidence is in the adult population. There are only a few cases found in the paediatric population [13,14].

## 2. Materials and Methods

Papers were found following a comprehensive search in PubMed, Google Scholar, and Scopus using the terms “cancer” or “malignancy” and “haemophilia” without time limits and using the English language as a filter. Indeed, we investigated a combination of the following terms: “cancer” or “malignancy” and “haemophilia” and “thrombotic risk” and “cancer” or “malignancy” and “haemophilia” and “paediatrics”. The references of all retrieved original articles and reviews were assessed for additional relevant articles.

## 3. Results and Discussion

The studies reviewed show strong heterogeneity regarding the population analysed and the type of tumour studied.

### 3.1. Relationship between Haemophilia and Cancer: Role of Plasma-Derived Clotting Factors

The in vitro and in vivo studies found that severe haemophilia protects against cancer development, but this is not confirmed for mild haemophilia. Patients with haemophilia may have an increased risk of developing certain types of cancer, such as liver cancer and Hodgkin’s disease, because of the chronic inflammation and cell damage that can occur due to repeated bleeding episodes and replacement therapy [15,16]. The history of haemophilia treatment is marked by challenges related to using plasma-derived products, particularly during the earlier years before more advanced treatment methods were implemented. Haemophilia patients, especially those treated with clotting factor concentrates derived from pooled human plasma, faced significant risks of contracting bloodborne infections such as HIV and HCV [16,17].

During the 1970s and 1980s, before the development of recombinant clotting factor concentrates, many haemophilia patients received treatment with plasma-derived products. Unfortunately, some of these plasma pools were contaminated with bloodborne pathogens. As a result, a significant number of haemophilia patients were infected with HIV and HCV, leading to serious health complications [8].

The contamination of clotting factors concentrated with HIV and HCV had important consequences for the haemophilia community. Many individuals developed AIDS (acquired immunodeficiency syndrome) due to HIV infection, and others suffered from chronic hepatitis C [18,19]. Tragically, some haemophilia patients also experienced an increased risk of cancer, often associated with these viral infections: non-Hodgkin’s lymphomas HIV-associated and hepatocellular carcinomas (HCC) HCV-associated [20,21,22].

Miesbach et al. discovered that older adults with HIV had a four times higher cancer prevalence than the age-matched general population, excluding hepatocellular carcinoma (HCC) [12]. There is an increase in HCC incidence in the overall haemophilia population, and most cases are due to HCV infection. As HCV infection is the most crucial risk factor for HCC in the haemophilia population, regular screening for liver disease is necessary, and a multidisciplinary approach is required to provide optimal therapeutic options [17]. However, there is no evidence of an increased incidence in other malignancies in patients with haemophilia compared to the general population [20]. The study also found that the prevalence of cancers among PWH was higher than in the general population, even after excluding patients with HIV or HCV infection [23]. While the hypothesis that haemophilia may offer protection against diseases such as cancer is intriguing, it remains unproven and needs further investigation [20]. The available data on the clinical management of cancer in haemophiliacs are limited, with most studies based on anecdotal case reports [24]. However, a retrospective survey conducted by the Association of Haemophilia Centres (AICE) found that non-virus-related cancers were less common in severe haemophilia patients than in those with milder forms of the disorder [16]. Haemophilia is not a direct risk factor for cancer, but the treatments for haemophilia can increase the risk. In particular, blood transfusions and clotting factor replacement therapy may be associated with increased cancer risk. This concern has been raised in previous research, which has suggested that exposure to blood products may increase the risk of viral infections that can contribute to cancer development [25]. Anticoagulants have been studied for their potential to prevent and treat cancer [26]. Some studies have suggested some characteristics of cancer incidences in patients treated with anticoagulants. Anticoagulants, such as warfarin, may have antitumour effects by inhibiting tumour invasion and preventing the breakdown of extracellular matrix protein [27]. Regarding experimental in vitro studies, the most important ones published to date are those conducted by Bruggemann et al. and Langer and colleagues. Both studies used a mouse model of Haemophilia A with a melanoma cell line capable of producing lung metastases. The first study showed that factor VIII replacement therapy in haemophilic mice increased the formation of lung metastases. In contrast, a direct thrombin inhibitor known as lepirudin inhibited lung seeding, indicating that thrombin generation contributed to pulmonary metastasis, even without factor VIII. The second study revealed that factor VIII deficiency reduced metastatic spread. Mice with factor V Leiden developed more metastases than wild-type controls, identifying endogenous thrombin as a significant factor in tumour dissemination [28,29].

### 3.2. Mortality in Patients with Cancer and Haemophilia

There are no studies in the literature that distinguish mortality due to bleeding from mortality due to disease progression in haemophiliacs with cancer.

Figure 1 shows the main literature data on cancer in haemophilic patients by geographical area.

#### 3.2.1. Mortality in Patients with Cancer and Haemophilia until the 1990s

Excluding viruses, the literature finds conflicting results on cancer mortality in haemophilia patients [17]. Except for the HCV associated hepatocellular carcinomas and HIV-associated lymphomas, mortality rates for cancer in mild haemophilia patients seem the same as in the general population [23]. However, Plug et al.’s research suggested that mortality was 2.3 times higher in haemophilia patients than in the general male population. Furthermore, HIV and hepatitis C largely influence the mortality of haemophilia patients. In patients with severe haemophilia not infected with viruses, mortality is 40% higher when compared with the general population [32]. Several studies have reported a higher prevalence of cancer in PWH than in the general population, where PWH is diagnosed with cancers at a much younger age [15,23,33,38,39,40]. However, there is limited data on the epidemiology of non-virus-related malignancy in PWH [12,41]. Darby et al. reported cause-specific mortality from 1977 to 1998 in males in the United Kingdom with Haemophilia A or B who were not infected with HIV compared with national mortality distinguished by severe and moderate/mild haemophile. The authors noted that mortality from liver cancer and Hodgkin’s disease was increased compared with mortality in the general population (standardized mortality ratio of 13.51 and 4.95, respectively). The standardized mortality ratio in severe cases is 4.98, and in moderate/mild cases, it is 4.95 [22].

Between 1970 and the early 1980s, mortality rates among PWH significantly decreased due to the widespread availability of clotting factor replacement products and the establishment of specialised haemophilia treatment centres (HTCs) that improved medical care [34]. The emergence of malignancies is one of the important causes of morbidity and mortality in PWH. In the past decade, the literature mainly focused on the epidemiology and outcome of blood-borne cancers in the haemophilia patient group, as the incidence of HBV, HCV, and HIV infections were high among them [29,31,37,41]. Triemstra and colleagues found no excess mortality from cancer in a cohort of Dutch haemophiliacs from 1986 to 1992 [35]. Darby et al. found that mortality from liver cancer was higher in haemophilic men in the UK treated with blood products contaminated with hepatitis C, with a 25-year cumulative risk of death from liver cancer of 0.37%, significantly higher than the expected rate of 0.03% from national mortality rates. The study also found that lymphoma cases in the UK haemophilia population, between 1978 and 1999, mainly occurred in HIV-positive patients [15]. In Canada, liver cancer or lymphoma deaths have increased significantly among HIV-positive individuals compared with HIV-negative individuals. Moreover, deaths due to all cancers were not increased in the population [36]. Soucie and colleagues found that a small percentage of deaths among haemophiliacs in six US states during 1993–1995 was attributable to non-HIV- or liver-related cancers with a standardised mortality ratio of 2.2 [34].

#### 3.2.2. Mortality in Patients with Cancer and Haemophilia after the 1990s

The challenges in haemophilia care have evolved. In the past, complications due to contaminated blood supply were the primary concern [18,26,36]. Indeed, with the availability of recombinant coagulation concentrates, age-related comorbidities such as malignancies, cardiovascular events, and diabetes have become important causes of morbidity and mortality [15,27,42,43]. Previous studies focused on blood-borne cancers, but more recent research has explored non-virus-related malignancies [1,12,44,45]. With the introduction of recombinant clotting factor concentrates (CFCs), physicians now pay attention to non-virus-related malignancies. Walker et al. reported that cancer-related deaths among HIV-negative PWH in Canada were lower than expected when it excluded liver cancer and lymphoma [36]. Furthermore, a systematic review by Miesbach et al. demonstrated that excluding HIV and hepatoma decreased the standard mortality rate for people with HIV [12]. While these studies provide valuable information, they are limited by small sample sizes. Additionally, differences in the timing of the studies could contribute to variations in cancer prevalence and outcomes, which the widespread introduction of CFCs and modern comprehensive haemophilia care may influence. Huang et al. conducted a nationwide population-based analysis of the occurrence and survival of cancer in PWH between 1997 and 2010 [23]. The study showed that PWH had a survival time similar to that of the general population after acquiring cancer. However, long-term treatment with CFCs can intensify thrombin-induced metastases, or the replacement dosage may be too low to generate such an effect [23]. The incidence and survival of cancers among PWH in Taiwan are increasingly affected by age-related diseases such as cancer. Huang et al. analyzed the data on 1054 PWH compared with 10,540 age- and gender-matched healthy individuals from the general population. The study found that PWH had a higher incidence of cancer, including hepatocellular carcinoma, than the general population. PWH who developed cancer were younger and had fewer comorbidities at diagnosis, but survival rates were as in the general population [23]. In particular, at the time of cancer diagnosis, PWH were 45.1 years compared to the general population that has a mean age of 57.2 years old (*p*  <  0.001). Moreover, also Biron-Andreani reported a median age at diagnosis of 54 (20–79) years [46]. During the study period, 19 PWH with cancers died, 24 patients were alive, and the median age at death was 47 years. No bleeding-related death was recorded among these deceased haemophilia patients. More recently, Darby and colleagues found that mortality from liver cancer and Hodgkin’s disease was substantially increased compared with mortality in general in the UK. There was no evidence of increased mortality from other cancers [15]. In 2021, Hassan et al. reported that survival in patients with haemophilia in the Netherlands has improved over time but is still lower than that of the general population. In particular, compared with the general Dutch male population, mortality of patients with haemophilia was still increased (SMR: 1.4, 95% confidence interval: 1.2–1.7) [30]. From 2001 to 2018, frequent causes of death were non-hepatic malignancies (26%) and intracranial bleeding (14%). Acquired immunodeficiency syndrome (AIDS; 2%), chronic liver disease (7%), and hepatocellular carcinoma (7%) were less frequent causes of death [30]. However, previously in the Netherlands, a prospective cohort study found that deaths from malignant neoplasms accounted for 22% of deaths in the Dutch haemophilia population from 1992 to 2001, with a standardised mortality ratio (SMR) of 1.5 showing a highly increased risk of death from hepatocellular carcinoma [32].

Table 1 reports of studies describing the standardized mortality ratio for hepatocellular carcinoma in haemophilia patients.

### 3.3. Haemophilia and Bleeding Risk in Cancer Patients 

Various factors influence the bleeding risk in patients with both haemophilia and cancer, and managing this population requires a careful and personalized approach. In general, cancer treatments can affect the production of blood cells and may increase the risk of bleeding in PWH. In addition, treatments such as radiotherapy damage blood vessels, which can lead to bleeding [15,33,39]. PWH undergoing cancer treatment may require adjustments to their replacement therapy to manage the increased risk of bleeding. While there is a potential link between bleeding and cancer, regular monitoring and appropriate management can help minimise the risks and ensure PWH outcomes. Bleeding and thrombotic complications are increasingly common in critically ill patients with cancer due to progress in cancer treatment and critical care. Bleeding infrequently occurs in patients with solid tumours, while leukaemias can lead to bleeding in up to 90% of patients [44,47,48,49]. Moreover, cancer patients with changes in haemostasis can lead to a prothrombotic state and venous thromboembolism [48]. The relationship between the coagulation system and cancer has been recognised since the late nineteenth century [50]. Evidence includes tumour cells within thrombi, thromboplastin activity and the production of procoagulant factors by tumour cells [51,52]. Cancer patients often have haemostasis alterations: platelet abnormalities, abnormal coagulation cascade activation, and decreased hepatic synthesis of anticoagulant and coagulant proteins [48]. Cancer can cause qualitative and quantitative platelet changes, including thrombocytosis and thrombocytopenia. Platelet activation by tumour cells generates thrombin, which stimulates platelets to supply factor V and the phospholipid necessary to activate factor X [53].

Cancer also causes an increase in coagulation factors and fibrinogen as increased levels of coagulation activation cause markers to indicate a consumptive coagulation process [44,54]. Various malignancies may secrete plasminogen activators, improving blood vessel wall permeability and promoting tumour cell migration [55]. Decreased hepatic synthesis of anticoagulant proteins correlates with serum albumin levels and is attributed to hepatic metastases and the effect of tumour necrosis factors [55,56]. As a consequence, we can state that the risk for bleeding or thrombosis in a patient with cancer depends on the underlying type of malignancy but also on the modality of anticancer therapy [57]. Factors contributing to the increased risk for bleeding and thrombotic complications in patients with cancer admitted to the intensive care unit (ICU) include indwelling catheters, systemic inflammatory response syndrome (SIRS), sepsis, prior chemotherapy and radiation treatment, and metastatic disease to the liver or bone marrow [48]. The association between haemostasis and cancer has been acknowledged for approximately 150 years [58].

Over time, several researchers have examined this relationship, particularly emphasising endogenous thrombin as a significant factor in tumour implantation, seeding, and metastasis [20,58,59]. Indeed, activated coagulation factors can trigger endothelial cells and platelets, releasing growth factors and the proliferation of tumours. The mechanism that leads to the formation of a complex involving cancer cells, platelets, and fibrin is particularly interesting. This complex protects cancer cells against mechanical stress and the host immune system, specifically natural killer cells. Furthermore, it promotes cancer-cell adhesion to the vascular endothelium and facilitates tumour-associated angiogenesis [60]. Cancer treatment is crucial for the risk of bleeding. Haemorrhagic complications occur more frequently in patients receiving chemotherapy or radiotherapy than in those undergoing invasive or surgical procedures, and continuous prophylaxis during these treatments is recommended. In most cases, haemophilia did not preclude access to chemotherapy or radiotherapy regimens [61]. Indeed, in 2007, Akl et al. revealed an increased bleeding risk in patients with cancer who had no therapeutic or prophylactic indication for oral anticoagulation [62]. Franchini et al. discuss the challenges of managing cancer in patients with haemophilia, particularly concerning the increased risk of bleeding during surgery or other invasive procedures [25]. The authors highlight that the close collaboration between haematologists and oncologists is essential in ensuring that cancer treatments are managed safely and effectively. Overall, the study emphasises the need to understand the potential links between haemophilia and cancer and improve management strategies for cancer in patients with haemophilia. The available literature shows that haemophilia can affect the treatment, clinical presentation, and diagnosis of neoplasia. However, only a few cases reporting on managing cancer in haemophilic patients exist. Studies have focused on managing hepatocellular carcinoma in haemophilic patients, which is a leading cause of death in haemophilic patients with chronic hepatitis C.

### 3.4. Prophylaxis and Treatment in Haemophilia Patients with Cancer

Franchini et al. highlight the importance of regular cancer screenings in people with haemophilia and suggest that more research is needed to better understand the relationship between haemophilia and cancer and to develop new treatment options tailored to the needs of patients with these conditions [25].

Prophylaxis in haemophilia and cancer patients is a complex issue and depends on several factors:Type of Haemophilia: The specific type of haemophilia (A or B) and the severity of the deficiency in the coagulation factor will influence the bleeding risk.Type of Cancer: The type of cancer can impact the risk of bleeding. Some tumours, especially those invading or damaging blood vessels, may increase the risk of bleeding.Cancer Treatments: Cancer treatments such as chemotherapy or radiation therapy can affect blood coagulation and elevate the risk of bleeding in patients with haemophilia.Surgical Interventions: Surgeries associated with cancer treatment can increase the bleeding risk in patients with haemophilia. Close coordination with a multidisciplinary medical team is essential to plan and manage these interventions safely.Monitoring and Management: PWH and cancer need close monitoring to promptly identify any signs of bleeding. Management often involves administering specific coagulation factors to correct the deficiency and prevent or treat bleeding.Collaboration Among Specialists: Managing patients with both haemophilia and cancer requires collaboration among specialists, including haematologists, oncologists, and other healthcare professionals. Planning a coordinated treatment plan is crucial to ensuring effective bleeding risk management.

Cancer-invasive diagnostic procedures and treatments should only be performed after the patient receives appropriate replacement therapy in close collaboration with haemophilia specialists. With advancing age, patients with mild haemophilia may develop malignancies that necessitate surgical or chemotherapeutic interventions. Moreover, chemotherapy-induced toxicity affecting blood cells, mucosal surfaces, or haemostasis can hinder the successful completion of anti-cancer treatments, sometimes requiring dose adjustments [25].

The increased bleeding risk associated with haemophilia must be considered when administering chemotherapy, radiotherapy, or invasive diagnostic procedures [25]. A case report by Lambert et al. demonstrated that chemotherapeutic agents targeted the vascular endothelium in mild Haemophilia B and colon cancer with liver metastases without exacerbating the bleeding diathesis [63]. Dawson et al. described the first autologous stem cell transplantation in a patient with severe Haemophilia A and HIV-related plasmablastic lymphoma of the oral cavity without bleeding complications [64]. Toyoda et al. analysed the safety and complications of interventional radiology for hepatocellular carcinoma in haemophilic patients [65]. They concluded that careful attention must be paid to gastrointestinal bleeding as a complication [65]. Shen et al. described a successful partial hepatectomy in a patient with haemophilia and hepatocellular carcinoma with no postoperative bleeding complications thanks to administering factor VIII concentrate during and immediately after surgery [66]. Doyle et al. evaluated the available HPT treatment options associated with haemophilia [67]. The authors collected 48 articles: 40 single case reports, seven single centre case series, and one multicentre case series [68,69,70,71,72,73,74,75,76,77,78,79,80,81,82,83,84]. They concluded that “timely surgical intervention with adequate haemodynamic support and the consideration of adjuvant therapies in selected cases can achieve acceptable outcomes in this cohort of patients”.

### 3.5. Thrombin in PWH and Cancer

Reduced production of thrombin, a potent cancer promoter, could potentially account for the lower incidence of cancer in haemophiliac patients [10]. After 30 years of monitoring, in Sweden, an elevated occurrence of cancer was identified in individuals with Haemophilia A or B, predominantly manifesting as haematological malignancies and urinary organ cancers [85]. The shared rise in cancer incidence in both Haemophilia A and B cases suggests a potential association with reduced thrombin generation rather than the specific factors FVIII or FIX themselves to malignancy [86]. Additionally, the inflammatory response following bleeding could contribute to the processes of carcinogenesis and metastasis. Understanding the intricate interplay between coagulation and the development or spread of tumours is challenging [87]. Despite the findings in the Swedish study for individuals with Haemophilia A or B, some preclinical indications propose a role for normal or above-normal thrombin generation in cancer metastasis. In a mouse model of Haemophilia A, lepirudin, a direct thrombin inhibitor, significantly reduced lung tumour seeding [29]. Furthermore, among patients with advanced malignancy, the anticoagulant dalteparin, a low-molecular-weight heparin, was observed to enhance survival in a subset with a more favourable prognosis [88,89].

Downstream of thrombin generation, the formation of a complex involving cancer cells, platelets, and fibrin is believed to promote the adhesion of cancer cells to vascular endothelium. This interaction then serves as a matrix for tumour-associated angiogenesis [29,60,90]. The complex relationship between coagulation and tumour development underscores the need for a comprehensive understanding of these processes.

### 3.6. Thrombosis in Patients with Haemophilia and Cancer 

The relationship between thrombosis and cancer was first recognized in 1823 by Bouillaud, and similar findings were extensively described by Armand Trousseau some years later [58,91,92]. Older PWH is at risk for developing ageing comorbidities, such as cardiovascular disease (CVD) and cancer [41]. Generally, there are hypotheses of a direct protective effect of haemophilia on developing ischaemic heart disease [93]. However, there are few cohort studies involving PWH with CVD and cancer, and the risk for these two age-related comorbidities in PWH is often left to the physician’s discretion [41]. Haemophilia, in theory, provides physiological protection against venous thromboembolism. Rare cases of thrombosis have been described in the literature among PWH [94]. In-dwelling catheter thrombosis is the most frequent cause in haemophilia patients [95]. Thrombosis has recently been described with novel non-replacement drugs [96]. Thrombosis can be a complication of major surgery in the case of intensive replacement therapy [97]. Cancer itself can promote thrombosis, which can add the risk of surgery or an indwelling catheter. In this case, haematologists and oncologists must evaluate the treatment very carefully: the therapy must consider the need to protect the patient from a progression of thrombosis on the one hand and, on the other, the haemorrhagic risk inherent in anticoagulant therapy.

## 4. Conclusions

As the life expectancy of PWH increases, more and more age-related diseases, such as cancer, emerge in this patient group. Generally, patients with haemophilia do not have an increased risk of developing cancer compared to the general population, but new studies with large case series are needed to confirm it; increased risk is found in the presence of HIV or hepatitis infection. Survival depends on age, early diagnosis, treatment choice, and management of bleeding disorders. It is important for PWH to be screened regularly and to coordinate with an expert team, including haematologists, oncologists, and other specialists, for optimal health management. It is also important that any invasive manoeuvre or chemotherapy treatment to be chosen that increases the risk of bleeding in a haemophilia patient with cancer is carefully evaluated by the oncologist and the specialist treating the patient with haemophilia.

Take home message: in addressing the co-occurrence of haemophilia and cancer, it is imperative to integrate personalised treatments and collaborative care models. By leveraging precision medicine and advancing interdisciplinary efforts, we can optimise management strategies, minimise complications, and enhance the overall well-being of patients facing these dual challenges. Early cancer detection is crucial because it offers a better prognosis and decreases mortality risk. In addition, disease management is generally less complex and more effective when the tumour is detected early. If patients who develop a malignancy are not on prophylaxis, it is recommended to pursue it as it would limit bleeding due to treatments and invasive manoeuvres.

## Figures and Tables

**Figure 1 jcm-13-01770-f001:**
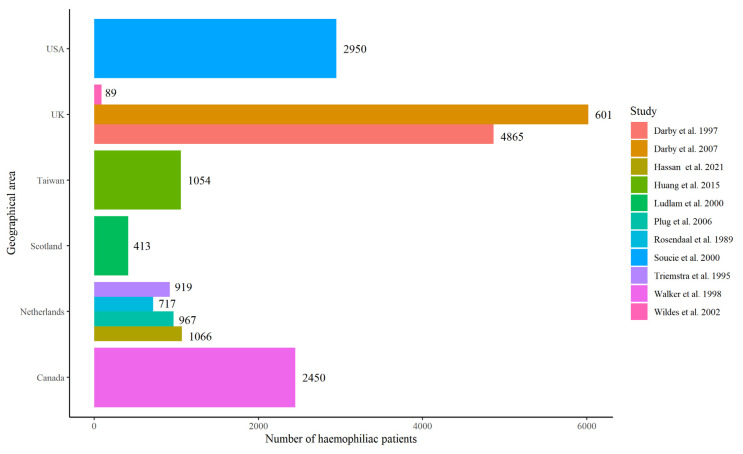
Summary of the main literature data on cancer-related deaths in haemophilic patients by geographical area. USA: Colorado, Georgia, Louisiana, Massachusetts, New York, and Oklahoma [15,22,23,30,31,32,33,34,35,36,37].

**Table 1 jcm-13-01770-t001:** Summary of studies that reported mortality estimates or deaths for hepatocellular carcinoma (HCC).

Study	Country and Study Period	Study Type	PWH	Severity N (%)	Age (Range)	Deaths for HCC	SMR (95% CI)
Darby et al. (2007) [15]	United Kingdom 1969–1992	Retrospective	4865	Mild and Moderate: 2641 (54.3) Severe: 2224 (45.7)	/	5	5.6 (1.8–13.0) for liver cancer
Plug et al. (2006) [32]	The Netherlands 1992–2001	Prospective	967 HA: 796 HB: 171	Mild: 414 (42.8) Moderate: 167 (17.3) Severe: 386 (39.9)	Mean: 52 (14–83)	5	17.2 (5.2–35.9)
Tagliaferri et al. (2012) [16]	Italy 2000–2007	Retrospective	3498	/	Median: 43 (3–88)	20	/
Lövdahl et al. (2013) [40]	Sweden 1968–2009	Retrospective	1432	Mild: 405 (43.4) Moderate: 145 (15.5) Severe: 384 (41.1) (No classification for remaining 497 PWH)	Mean: 62.7	/	/

PWH = Patients With Haemophilia; SMR = Standardized Mortality Ratio/ = no data.

## Data Availability

The data presented in this study are available on request from the corresponding author.

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
