# Peer review of "Haemophilia and Cancer: A Literature Review"

_jcm, 2024, doi:10.3390/jcm13061770_

Round 1
Reviewer 1 Report
Comments and Suggestions for Authors
Zanon et al. review the correlation between haemophilia and cancer comprehensively. This paper is very informative and worth publishing.
There needs to be some minor revisions as follows.
1. Using only “haemophilia” against “hemophilia” would be better.
2. On page 3, lines 101-107, from “Souncie et al. in 2000 ...” to “... in non-viral malignancies is contradictory.”, I can’t get clearly what you want to say, and the flow is difficult to understand.
3. On page 3, line 111, “Because hepatocellular carcinoma (HCC) appears...” can be changed to “Because HCC appears...” since HCC already appeared on line 104.
4. In Table 1, Does “/” means “no data”? Please clarify in this table.
5. On page 5, line 195, “The second study revealed ( that” should revised to “The second study revealed that”.
6. On page 6, line 259, “HPT” needs to be spelled out.
7. On page 6, lines 272-274, it is better to show the example of a disease like “such as leukemia”.
Author Response
Dear reviewer , thank you very much for taking the time to review this manuscript. I' sending you the minor revision requested ( please see the attachment) Best regards Ezio Zanon
Zanon et al. review the correlation between haemophilia and cancer comprehensively. This paper is very informative and worth publishing.
There needs to be some minor revisions as follows.
- Using only “haemophilia” against “hemophilia” would be better.
Dear Reviewer, we have made all the changes.
- On page 3, lines 101-107, from “Souncie et al. in 2000 ...” to “... in non-viral malignancies is contradictory.”, I can’t get clearly what you want to say, and the flow is difficult to understand.
Dear reviewer, we have restructured the entire paragraph.
- On page 3, line 111, “Because hepatocellular carcinoma (HCC) appears...” can be changed to “Because HCC appears...” since HCC already appeared on line 104.
We fixed it.
- In Table 1, Does “/” means “no data”? Please clarify in this table.
We add the meaning below the table.
- On page 5, line 195, “The second study revealed ( that” should revised to “The second study revealed that”.
We fixed it.
- On page 6, line 259, “HPT” needs to be spelled out.
We fixed it.
- On page 6, lines 272-274, it is better to show the example of a disease like “such as leukemia”.
We replaced “such as” instead of “like”.

Reviewer 2 Report
Comments and Suggestions for Authors
This is unfortunately a rather confusing review about cancer and hemophilia.
My considerations concern:
The structure of the manuscript should be reviewed (large paragraphs, references in the middle of the sentences etc)
Introduction should be enriched with data about hemophilia. Thrombin generation is associated with cancer and metastasis, a fact that should be commented.
It is well known that hemophilia patients suffered from HIV and HCV related cancer due to plasma derived products, that was used in the past. Further description is needed.
Lack of thrombin generation was thought to be the anti-cancer protective mechanism in hemophilia patients. In the era of widespread prophylaxis is this protective mechanism missing? Is there a difference in cancer prevalence between severe and mild hemophilia?
What is known about pediatric population and cancer?
Acquired hemophilia should be reported separately.
Comments on the Quality of English LanguageExtensive editing required
Author Response
Dear reviewer, thank you very much for taking the time to review this manuscript. Please find the detailed responses below . I have extensively revised the text according to your suggestions. I checked the English with Grammarly software.
Best regards
Ezio Zanon
checked the English with Grammarly software.
This is unfortunately a rather confusing review about cancer and hemophilia.
My considerations concern:
The structure of the manuscript should be reviewed (large paragraphs, references in the middle of the sentences etc).
Dear Reviewer, we have restructured the first paragraphs and also added on the advice of the Editor, a figure. In addition, we put all bibliographical citations at the end of the sentence.
Introduction should be enriched with data about hemophilia. Thrombin generation is associated with cancer and metastasis, a fact that should be commented.
Dear reviewer. We have added an introductory section on haemophilia. Indeed, we added a specific paragraph for thrombin, cancer and hemophilia. Thank you for your suggestion.
It is well known that hemophilia patients suffered from HIV and HCV related cancer due to plasma derived products, that was used in the past. Further description is needed.
We have added a few more sentences of explanation about treatments and the influence of viruses in the Relationship Between Haemophilia and Cancer section
Lack of thrombin generation was thought to be the anti-cancer protective mechanism in hemophilia patients. In the era of widespread prophylaxis is this protective mechanism missing? Is there a difference in cancer prevalence between severe and mild hemophilia?
We have inserted a new paragraph on the subject.
What is known about pediatric population and cancer?
We found no articles on the paediatric population except for a single case with sarcoma, which we cite.
Acquired hemophilia should be reported separately.
We have not mentioned acquired haemophilia except in one sentence, which we have removed.
